# Health and wellness in the Australian coal mining industry: A cross sectional analysis of baseline findings from the RESHAPE workplace wellness program

**Aaron Bezzina**[1,2]*, **Emma K. Austin**[1], **Trent Watson**[2,3], **Lee Ashton**[2,4], **Carole L. James**[1,2]

**1** Centre for Resources Health and Safety, College of Health, Medicine and Wellbeing, University of Newcastle, Callaghan, Australia, **2** School of Health Sciences, College of Health, Medicine and Wellbeing, University of Newcastle, Callaghan, Australia, **3** Ethos Health, Newcastle West, Australia, **4** Priority Research Centre for Physical Activity and Nutrition, University of Newcastle, Callaghan, Australia

* Aaron.bezzina@newcastle.edu.au

**Data Availability Statement:** Data cannot be shared publicly because of confidentiality reasons. Data are available from the University of Newcastle

## Abstract

Overweight and obesity has reach pandemic levels, with two-thirds (67%) of adult Australians classified as overweight or obese. As two of the most significant behavioral risk factors for obesity are modifiable (diet and exercise), there exists an opportunity for treatment through workplace health promotion initiatives. As one of Australia's largest industries with its own unique workplace factors, the mining industry has previously reported higher than population levels of overweight and obesity. This represented an opportune setting to test the RESHAPE workplace wellness program. RESHAPE is an eight-step framework (based on the *WHO 'Health Workplace Framework and Model')* which aims to provide a sustained approach to wellness in the workplace. This paper presents baseline findings from a pilot study that aimed to implement RESHAPE at three mine sites in NSW, Australia, and investigates the issue of overweight and obesity in the coal mining industry. Across three mine sites, 949 coal miners were examined cross-sectionally on a range of workplace, wellness, health, diet, and exercise factors using a paper-based survey. This was a predominantly male sample (90.4%) with the majority (59.2%) of participants aged 25–44 years. Self-reported height and weight measures indicated that less than 20 percent (18.9%) of participants were in a healthy BMI range, while there were effectively equal numbers of overweight (40.9%) and obese (39.1%) participants. Only 3.5% of participants met the daily recommendation for vegetables (5 serves) and shift-workers had greater association with elevated BMI compared to non-shift workers (B = 1.21, 95% CI: 0.23, 2.20, p = 0.016). Poor nutrition is likely to be a key component in elevated levels of overweight and obesity within this industry, with workplace factors compounding challenges workers face in implementing health behavior change. Future studies would benefit from assessing diet and physical activity knowledge in relation to recommendations and serving sizes.

Institutional Ethics Committee (contact via human-ethics@newcastle.edu.au) for researchers who meet the criteria for access to confidential data.

**Funding:** This research was supported by competitive grant funding from Coal Services Health and Safety Trust G1800980. This grant provided funding for the salary for author EA. The funding body did not have any additional role in the study design, data collection and analysis, decision to publish, or preparation of the manuscript. All other salaries were provided as in-kind support either from the University of Newcastle (AB, LA, CJ) or commercial entity Ethos Health (TW). The specific roles of these authors are articulated in the 'author contributions' section.

**Competing interests:** The authors would like to acknowledge the commercial affiliation of author TW and funding received from Coal Services Health and Safety Trust. Author TW as a commercial affiliate did not receive any funding for his consultancy on this research, and provided all support in kind in his supervisory role for PhD candidate and lead author AB. The commercial affiliation and funder do not alter our adherence to PLOS ONE policies on sharing data and materials. The authors have no other competing interests to declare.

## Introduction

Overweight and obesity has reached pandemic levels, with an estimated 1.9 billion adults globally classified as overweight, and of these, 650 million obese [1]. Global projections show no sign of slowing down, with an estimated 2.16 billion adults to be classified as overweight and a further 1.12 billion obese by 2030 [2]. In particular, Australia currently ranks within the top 10 of all Organisation for Economic Co-operation and Development (OECD) countries in terms of obesity rates as a percentage of total adult population [3]. Results from the 2017–18 Australian National Health Survey show that over two thirds (67%) of the adult population were classified as either overweight or obese (12.5 million people), up from 63.4% in the previous survey in 2014–2015 [4]. This change represents an increase in the proportion of adults classified as obese, rising from 27.9% to 31.3% in just three years [4].

Obesity implicates many aspects of health, playing an antagonistic role in the development of a range of non-communicable diseases including type 2 diabetes, fatty liver disease, cardiovascular disease, and certain cancers [5]. The multifactorial nature of obesity makes treatment complicated [5]. In spite of this, the two most significant behavioural risk factors, diet and exercise [6], are modifiable and represent an opportunity for treatment through health promotion initiatives [7]. The workplace is an ideal setting for the delivery of health promotion programs as employees typically spend one-third of their waking lives at work [8]. In conjunction with time spent at work, the workplace offers the necessary infrastructure to support large scale health promotion [9]. The health and wellbeing of workers is vital to ensure workplace safety, as well as productivity, as it is related to absenteeism and presenteeism, and underpins many psychosocial workplace factors as well as quality of life [10].

As one of Australia's largest industries, with 247,300 employees [11], the mining sector contributed significant export earnings of $290 billion in 2019–2020, which is approximately 9% of Australia's Gross Domestic Product [12]. The mining industry plays a significant role in rural and remote areas by generating jobs and infrastructure and contributes to both local and national economies. Rates of overweight and obesity in New South Wales (NSW) male coal miners have been shown to be considerably worse than national figures, with 84.9% of males overweight or obese [13] compared to 74.5% nationally (females tracking evenly at 58.0% and 59.7% respectively) [4]. This aligns with men accounting for more than 90% of the NSW mining workforce [13], and the population prevalence of overweight and obesity being greater in men [4].

The Australian mining sector maintains a strong emphasis on safety however there has been less emphasis on health as part of workplace health and safety until recent years. One of these recent developments was the: *'Blueprint for the management of overweight and obesity in the NSW mining industry'* which was developed in response to an industry identified need and stakeholder forum [14]. The blueprint highlighted the need for RESHAPE, which was conceived by a sub-committee of the New South Wales Minerals Council (NSWMC) Work Health Safety Committee. This committee developed the framework for a coordinated plan of action for mining operations to engage in industry-wide change to prevent and manage obesity. RESHAPE is an eight-step framework which aims to provide a sustained approach to wellness in the workplace.

The mining industry faces many unique occupational factors which promote higher BMI values compared to other labour industries. The sector is heavily male dominated with higher than average rates of obesity found in men [4]; mining jobs often utilise shift work which has been identified as a risk factor for excessive weight [15]; as well as mining jobs are often found in regional and remote areas which experience greater obesity rates compared to metropolitan

areas. All of these factors underscore the need for RESHAPE and a coordinated approach to wellness in this industry.

Considering the aforementioned factors, this study aims to implement the RESHAPE framework into three Australian coal mining sites. It is hypothesised that the RESHAPE framework will lead to greater wellness outcomes for the individual (weight loss) compared to standard, once off wellness initiatives which are current practice in this industry. This paper presents baseline findings from a pilot study that implements RESHAPE and investigates the issue of overweight and obesity in the coal mining industry. Knowledge gaps remain alongside concerns regarding the success of diet and physical activity programs within blue-collar workers, particularly miners. This paper looks to bolster this shortcoming of the literature base and provides discourse concerning wellness in blue-collar male dominated industries.

## Materials and methods

### What is RESHAPE?

RESHAPE is an eight-step framework based on the *WHO 'Health Workplace Framework and Model'* [16]. Each of the eight-steps is accomplished by actioning several key activities to achieve the outcome of the step. By way of example, *Step 1*: *Mobilize*, involves Action 1: Identify key stakeholders, Action 2: Develop and adopt a RESHAPE charter, and Action 3: Align RESHAPE with organization practices.

Coal mining, and mining more generally, provides an incredibly dynamic and variable environment. RESHAPE does not adopt the mantra of one size fits all, which is novel within itself, rather it looks at problems pragmatically, and applies appropriate mine site specific solutions. Whilst this study is focusing on the issue of overweight and obesity within the workplace, the RESHAPE process is designed to be used for any modifiable health risk factor including smoking, nutrition, alcohol, physical activity, sleep or mental health.

The objective of RESHAPE is to provide a sustained approach to fostering healthy, happy, safe, and productive workplaces through sharing responsibility to create an environment and culture where healthy choices are the valued and easy choice. This produces organizational change through ongoing investment into workplace health, and ultimately cultivates a workplace culture which is conducive to positive change. The framework is championed by a working party and planning is pivotal, with 1, 3- and 5-year milestones driving future thinking. It must be stated that RESHAPE is not a specific program or a one-off intervention, rather an overarching framework for ongoing action and continuous improvement within the workplace. The framework has been implemented at sites across Australia and anecdotally demonstrated success, however, has yet to be executed and evaluated within an empirical setting which this study aims to address.

### Ethics

The project was approved by the University of Newcastle's Human Research Ethics Committee (approval number H-2019-0087).

### Study design

This study utilizes a quasi-experimental pre-test–post-test design, whereby the study participants across the three mines will be measured at baseline and again 12 months later (after completion of the workplace wellness intervention). This paper reports cross-sectional baseline results only.

## Recruitment

Three NSW mine sites, two open cut and one underground, were recruited to participate in the pilot study via a convenience sample of mine sites that expressed interest in participation. At the time of study inception, research members were chairing an obesity subcommittee with key stake holders (NSW Minerals Council and various mine sites across NSW) which had been formed in response to recommendations from the '*Blueprint for the management of overweight and obesity in the NSW mining industry*' report [14]. During one of these meetings, researchers sought expression of interest from 12 mine sites as well as other stakeholders who had previously conducted similar research projects with the university. Sites that expressed initial interest (n = 6) were sent an information package detailing the RESHAPE process and philosophy. Three sites joined the study, from the Hunter and Central West regions of NSW, Australia. The study was capped at three sites for funding reasons and a combination of underground and open cut was elected to capture different workplace characteristics.

Each site was supported by researchers to establish a working party to champion and action the RESHAPE ethos. At each mine sites' Occupational Health and Safety (OH&S) meetings, the study was discussed, and questions answered. The RESHAPE manual was provided which outlined the process of selecting working party members. The OH&S committee members decided who they wanted to recruit to their parties, how often they would meet, and other key considerations outlined in the RESHAPE manual. OH&S representatives were commonly on the working party.

## Baseline surveys

Paper based surveys were conducted during August–December 2019. At two sites, baseline surveys were conducted at training days and a member of the research team was present to introduce the study and answer worker questions. For logistical reasons the third site administered the surveys during pre-start meetings independently.

An information statement attached to the survey explained the aims of the project, the voluntary nature of the research and the confidentiality of data collected. In an attempt to link responses from baseline to follow-up, participants were asked to enter their surname and year of birth to generate a unique study code. Consent was obtained via a tick box at the end of the information statement. The expected completion time of the survey was 15–20 minutes.

## Measures

The baseline survey consisted of primarily multiple-choice questions, with a few open-ended responses. The survey included a range of measures, including workplace productivity [17], sleep and fatigue [18], and mental health [19]. In spite of this, these were not the focus of the study and are not reported in this paper. Quantitative measures related to personal and workplace characteristics, general health, physical activity and nutrition are detailed below.

## Personal characteristics

Personal characteristic questions were a mix of closed and open-ended responses. Occupation was open ended, however during data aggregation, this was grouped into four categories (production, trade and engineers, manager and other). Highest qualification completed ranged from: no formal qualification to higher university degree.

## Current work situation

Work situation questions pertained to employment status, usual hours worked in a week, shift-work status and hour worked per shift. All responses were close ended, excluding the usual hours worked per shift which invited participates to specify the number of hours when they exceeded 12.

## Health

Health questions were related to body anthropometrics (self-reported weight in kg and height in cm). BMI was calculated during data analysis from participant responses. Perception of weight status (do you consider yourself an acceptable weight, underweight or overweight), weight status history over the previous year, perception of general health (a five-response matrix ranging from excellent to poor), and health goals were also noted.

## Smoking

Smoking status and dependence was assessed via the Fagerstrom test for nicotine dependence containing the follow questions [20]: A dichotomous smoking status question, a four-response category question concerning cigarettes smoked per day (10 or less, 11–20, 21–30, 31 or more), and another dichotomous question concerning if the participant in the previous 12 months has tried to stop smoking.

## Alcohol

The short form Alcohol Use Disorders Identification Test (AUDIT) was used to determine alcohol dependency [21], and accompanying the questionnaire was a standard drinking guide infographic. Questions pertained to drinking frequency (five response question from never to four or more times a week), standards consumed during a drinking session (six response question ranging from 1–2 drinks to 10 or more drinks), and frequency of consuming six or more standard drinks on one occasion (five response question ranging from never to daily or almost daily).

## Nutrition

Nutrition questions were based on two different tools. Fruit and vegetable intake were quantified via an eight-response matrix question ranging from: "I never eat fruit/vegetables" to "I eat 6 or more serves per day". These questions were based on the Australian National Health Survey 2017–18 [4] and were validated for use in a general Australian population [22]. Infographics from the Australian Guide to Healthy Eating depicting correct serving sizes were also shown to support accurate responses. Frequency of sugar sweetened beverages (SSB) intake, discretionary takeaway meals and discretionary (fried) potato products were all quantified via nine response question matrixes, ranging from: less than 1 per month to more than 7 times per week. Meal portion sizes for the main components of a meal (potato, vegetables, steak and meat or vegetable casserole) were also quantified via an eight- question response matrix with pictures for each component (depicting increasing portion sizes). SSB, discretionary takeaway meals, fried potato products and portion sizes were all adapted from the Dietary Questionnaire for Epidemiological Studies (DQES) tool [23], which has been validated for use in Australian young adults [24].

## Exercise

Exercise questions were adapted from the Active Australian Survey [25], however, whilst being validated in mid-age women [26] and older adults [27], has yet to be validated in a general population setting. Total walking for fitness, recreation, or sport; total walking to get from place to place; total moderate exercise and total vigorous exercise per week were quantified via an open-ended question pertaining to minutes and hours of each activity per week. Results were transformed to a total minute sum, and stratified into the categories: 0 minutes, between 1 and 149 minutes, between 150 and 300 minutes and more than 300 minutes. This was done to allow comparison between our study participants and generalized Australian data reported in the National Health Survey 2017–18 [4].

## Data analysis

Survey data was cleaned, aggregated and transformed prior to analysis in SPSS [28]. Chi-square tests were used to test the relationship between categorical variables. A range of weight, diet, physical activity, and fatigue variables were tested against mine type, age, gender, and employment status.

Assumptions for regression models were checked before final interpretation. Linearity was assessed using scatter plots which showed a strong linear relationship between BMI and outcome variables. Normality was assessed via a histogram graph fitted with a goodness-of-fit test which showed normal distribution. Homoscedasticity was assessed via plot residuals against fitted values which showed minimal deviation from the standard. Lastly independence was assessed via plot residuals which showed random sequence.

# Results

## Study population

Across the three sites there were 949 participants at baseline (181, 347, 421 respectively at each site). Response rate of the survey differed between sites and collection methods; Site 1: 77.8%; Site 2: 69.9%; Site 3: Unknown. As described earlier during baseline survey methods, site 3 undertook a different collection procedure which meant response rates could not be tracked. The demographic characteristics of the study population are shown in Table 1. This was a predominantly male sample (90.4%) with the majority (59.2%) of participants aged 25–44 years. Self-reported height and weight measures indicated that less than 20 percent (18.9%) of participants were in a healthy BMI range, while there were effectively equal numbers of overweight (40.9%) and obese (39.1%) participants. Regarding highest level of education obtained, most participants had a trade/apprentices (37.8%) followed by year 10 equivalent (22.1%) and then certificate/diploma (17.6%). Participants were predominantly widowed/divorced/separated/or in a relationship but not living together (77.8%), were permanent full-time workers (78.0%), worked 39–45 hours a week (48.3%), were shift workers (80.1%) and were either working in production (i.e., as a coal miner) (60.2%) or as a trade/engineer (31.3%).

Table 2 outlines the results of the linear regression model for participant demographics and workplace factors with BMI as the effect. Those participants who worked 46–56 hours showed negative effect with BMI (B = -1.93, 95% CI: -3.83, -0.34, p = 0.46). Additionally, shift-workers had greater association with elevated BMI compared to non-shift workers (B = 1.21, 95% CI: 0.23, 2.20, p = 0.016). No other demographic or workplace factors were significantly associated with BMI.

Table 3 outlines the results of the linear regression model for participant diet questions with BMI as the effect. There were no statistically significant relationships between participants'

**Table 1. Baseline demographic characteristics of study cohort across all three sites.**

| Characteristic | n | % |
|---|---|---|
| *Gender* | | |
| Male | 845 | 89.8 |
| Female | 90 | 9.6 |
| Other | 2 | 0.2 |
| Prefer not to say | 4 | 0.4 |
| *Age* | | |
| 18–24 | 58 | 7.0 |
| 25–34 | 262 | 31.8 |
| 35–44 | 226 | 27.4 |
| 45–54 | 197 | 23.9 |
| 55–64 | 75 | 9.1 |
| 65–74 | 7 | 0.8 |
| *BMI Category* | | |
| Underweight | 9 | 1.1 |
| Healthy | 155 | 19.1 |
| Overweight | 335 | 41.3 |
| Obesity | 313 | 38.5 |
| **Smoking status** | | |
| Yes | 130 | 13.7 |
| No | 794 | 83.7 |
| **Alcohol consumption** | | |
| Never | 57 | 6.0 |
| Monthly or less | 152 | 16.0 |
| 2–4 times a month | 249 | 26.2 |
| 2–3 times a week | 310 | 32.7 |
| 4 or more times a week | 166 | 17.5 |
| *Physical activity* | | |
| 0 minutes | 73 | 17.3 |
| 1–149 minutes | 98 | 23.2 |
| 150–300 minutes | 94 | 22.1 |
| 300 + minutes | 158 | 37.4 |
| *Highest Qualification* | | |
| No formal qualification | 33 | 3.5 |
| Year 10 equivalent | 207 | 22.1 |
| Year 12 equivalent | 107 | 11.4 |
| Trade/Apprenticeship | 355 | 37.8 |
| Certificate/Diploma | 165 | 17.6 |
| University degree | 48 | 5.1 |
| Higher university degree | 18 | 1.9 |
| Prefer not to say | 5 | 0.5 |
| *Marital status* | | |
| Single | 129 | 13.8 |
| Widowed/Divorced/Separated/In a relationship but not living together | 728 | 77.8 |
| Married/Defacto/Living together | 64 | 6.8 |
| Prefer not to say | 15 | 1.6 |
| *Employment status* | | |
| Permanent full-time | 723 | 78.0 |

(*Continued*)

**Table 1.** (Continued)

| Characteristic | n | % |
|---|---|---|
| Permanent part-time | 10 | 1.1 |
| Fixed term | 22 | 2.4 |
| Casual | 37 | 4.0 |
| Contractor | 104 | 11.2 |
| Apprentice/Trainee | 31 | 3.3 |
| *Hours worked per week* | | |
| <24 | 4 | 0.4 |
| 24–38 | 195 | 21.2 |
| 39–45 | 443 | 48.3 |
| 46–56 | 227 | 24.7 |
| >56 | 49 | 5.3 |
| *Work shiftwork* | | |
| Yes | 744 | 80.1 |
| No | 185 | 19.9 |
| *Occupation Category* | | |
| Production | 571 | 60.2 |
| Trade and engineers | 297 | 31.3 |
| Manager | 25 | 2.6 |
| Other | 24 | 2.5 |

daily fruit or vegetable consumption and BMI. There was a statistically significant relationship with BMI and the increased daily consumption of SSB, with four cups per day (B = 4.24, 95% CI: 1.25, 7.23, p = 0.005) and seven cups per day (B = 6.82, 95% CI: 0.59, 13.1, p = 0.03) associated with higher BMI. Fast food consumption was also associated with BMI, with less than once a month (B = 2.81, 95% CI: 0.21, 5.43, p = 0.034), once a week (B = 3.47, 95% CI: 0.82, 6.12, p = 0.034), twice a week (B = 3.2, 95% CI: 0.27, 6.14, p = 0.032), four times a week (B = 6.45, 95% CI: 2.16, 10.74, p = 0.003) all illustrating a significant relationship, with more than seven times a week approaching significance (B = 11.11, 95% CI: -0.56, 22.78, p = 0.062).

Hot chip consumption showed a negative relationship with BMI, with higher consumption indicative of lower BMI (more than seven times a week: B = -6.14, 95% CI: -12.29,0.002, p = 0.05). Consumption of a sweet cake's once a week had a negative association with BMI (B = -2.18, 95% CI: -3.90, -0.46, p = 0.013).

Table 4 illustrates the results from the linear regression model regarding participants' physical activity with BMI as the effect. There were no significant relationships with walking for recreation, total walking excluding for physical fitness, total moderate physical activity and total vigorous activity, across all categories (0 minutes, between 1 and 149 minutes, between 150 and 300, more than 300 minutes per week). Similarly, there was no significant relationships between sitting time at work and BMI. Interestingly total physical activity minutes in the last week showed a significant relationship between 150-to-300-minute category and increased BMI (B = 1.81, 95% CI: 0.139, 3.485, p = .034).

Table 5 outlines the results from the multiple linear regression to predict BMI from shift work status, SSB consumption and fast-food intake. Whilst two of the variables (SSB and fast-food consumption) were not significant, shift work status was (B = 1.15, 95% CI: 0.15, 2.15, p = 0.025).

Fig 1 depicts a comparison of daily fruit consumption in serves per day between study participants and that of the general Australian population. General Australian population data

**Table 2.** Linear regression model demographic and workplace factors and the effects on BMI.

| Parameter | Co-efficient | Std. Error | 95% Confidence Interval | | P |
|---|---|---|---|---|---|
| | | | Lower | Upper | |
| **Age (years)** | | | | | |
| 18–24 | -1.056 | 2.2837 | -5.532 | 3.420 | .644 |
| 25–34 | 1.018 | 2.1785 | -3.252 | 5.287 | .640 |
| 35–44 | 1.504 | 2.1844 | -2.777 | 5.785 | .491 |
| 45–54 | 2.480 | 2.1901 | -1.812 | 6.773 | .257 |
| 55–64 | 2.183 | 2.2722 | -2.271 | 6.636 | .337 |
| 64–74 | Ref | Ref | Ref | Ref | Ref |
| **Gender** | | | | | |
| Male | -4.583 | 5.5811 | -15.522 | 6.355 | .412 |
| Female | -7.196 | 5.6084 | -18.189 | 3.796 | .199 |
| Other | 4.417 | 7.8322 | -10.934 | 19.768 | .573 |
| Prefer not to say | Ref | Ref | Ref | Ref | Ref |
| **Smoking status** | | | | | |
| Yes | .390 | .5975 | -.781 | 1.561 | .514 |
| No | Ref | Ref | Ref | Ref | Ref |
| **Alcohol consumption** | | | | | |
| Never | .426 | .9880 | -1.510 | 2.363 | .666 |
| Monthly or less | -.111 | .7102 | -1.502 | 1.281 | .876 |
| 2–4 times a month | -.789 | .6456 | -2.055 | .476 | .221 |
| 2–3 times a week | -.176 | .6177 | -1.386 | 1.035 | .776 |
| 4 or more times a week | Ref | Ref | Ref | Ref | Ref |
| **Highest qualification completed** | | | | | |
| No formal qualification | 2.015 | 3.5364 | -4.917 | 8.946 | .569 |
| Year 10 equivalent | -1.035 | 3.3155 | -7.533 | 5.463 | .755 |
| Year 12 equivalent | -.670 | 3.3476 | -7.231 | 5.891 | .841 |
| Trade/Apprenticeship | -1.057 | 3.3073 | -7.539 | 5.425 | .749 |
| Certificate/Diploma | -1.105 | 3.3277 | -7.628 | 5.417 | .740 |
| University degree | -3.245 | 3.4134 | -9.935 | 3.445 | .342 |
| Higher university degree | -1.164 | 3.5811 | -8.183 | 5.855 | .745 |
| Prefer not to say | Ref | Ref | Ref | Ref | Ref |
| **Employment status** | | | | | |
| Permanent full-time | -1.027 | 1.2223 | -3.423 | 1.369 | .401 |
| Permanent part-time | .920 | 2.2958 | -3.579 | 5.420 | .688 |
| Fixed term | -2.224 | 1.7024 | -5.560 | 1.113 | .191 |
| Casual | .264 | 1.5806 | -2.834 | 3.362 | .868 |
| Contractor | .001 | 1.3351 | -2.615 | 2.618 | .999 |
| Apprentice/Trainee | Ref | Ref | Ref | Ref | Ref |
| **Relationship status** | | | | | |
| Single | -1.321 | 1.8663 | -4.979 | 2.336 | .479 |
| Widowed/Divorced/Separated/In a relationship but not living together | -1.173 | 1.7950 | -4.691 | 2.346 | .514 |
| Married/Defacto/Living together | -2.311 | 1.9474 | -6.128 | 1.506 | .235 |
| Prefer not to say | Ref | Ref | Ref | Ref | Ref |
| **Hours worked each week** | | | | | |
| <24 | -3.023 | 3.4321 | -9.750 | 3.704 | .378 |
| 24–38 | -1.117 | .9805 | -3.038 | .805 | .255 |
| 39–45 | -1.683 | .9239 | -3.494 | .128 | .069 |

(*Continued*)

**Table 2.** (Continued)

| Parameter | Co-efficient | Std. Error | 95% Confidence Interval | | P |
|---|---|---|---|---|---|
| | | | Lower | Upper | |
| 46–56 | -1.933 | .9688 | -3.832 | -.034 | .046 |
| >56 | Ref | Ref | Ref | Ref | Ref |
| **Do you work shift work** | | | | | |
| Yes | 1.214 | .5019 | .230 | 2.197 | 0.016 |
| No | Ref | Ref | Ref | Ref | Ref |
| **Occupation category** | | | | | |
| Production | 2.120 | 1.3882 | -.601 | 4.841 | .127 |
| Trade and engineers | 1.306 | 1.3847 | -1.408 | 4.020 | .346 |
| Manager | 1.373 | 1.7560 | -2.069 | 4.815 | .434 |
| Other | Ref | Ref | Ref | Ref | Ref |

* Indicates significance at alpha level 0.05.

was obtained from the 2017–18 National Health Survey [4]. Results illustrate that only 42.2% of participants are meeting the requirement of at least two serves of fruit a day compared to 51.4% of adult Australians.

Fig 2 depicts a comparison of daily vegetable consumption in serves per day between participants and the general Australian population, taken from the 2017–18 National Health Survey [4]. The results show that only 3.5% of participants met the recommendation of a minimum of five serves of vegetables daily compared to 9.6% of adult Australians.

Fig 3 depicts a comparison of total physical activity in the last week between participants and the general Australian population, taken from the 2017–18 National Health Survey [4]. Participants in the study faired similar to the general Australian population with roughly 60% completing 150 minutes of physical activity or more over the week.

## Discussion

This paper reports the baseline results from three coal mine sites located in NSW, Australia, who were recruited to participate in a workplace wellness program. These results illustrate the systemic nature of overweight and obesity within the NSW coal mining industry; and identify potential factors, whether it be workplace, diet or exercise that may compound this issue. Whilst the results are alarming, this trend has been previously documented both in the Australian mining industry [13, 29], as well as other blue-collar male dominated employment sectors (construction and manufacturing) in Australia [30, 31].

Baseline results showed that nearly four out of five surveyed employees were either overweight or obese (41.3% and 38.5% respectively). Comparably, to NSW mining industry health data from 2014, this is a rise of over 5.3% in terms of cumulative population exceeding a healthy BMI [13]. Whilst the problem of overweight and obesity is more pronounced in this industry, the progressive increase in BMI also mirrors that of national figures. From the period of 2014–15 to 2017–18, within Australia, overweight and obesity increased from 63.4% to 67%, which was mainly driven by rise of adults classified as obese (27.9% to 31.3% respectively) [4]. Moreover, our results highlight that a large proportion of study participants were classified as obese (38.5%). Juxtaposed to a comparable dataset of NSW mining industry employees from 2014 which reported obesity rates of 27.1% [13], this represents a rise of 11.4% in the proportion of employees classified as obese.

**Table 3. Relationship between participant self-reported diet and physical activity responses and the effect on BMI.**

| Parameter | Co-efficient | Std. Error | 95% Confidence Interval | | P |
|---|---|---|---|---|---|
| | | | Lower | Upper | |
| **Serves of fruit per day** | | | | | |
| I never eat fruit | -4.235 | 3.5673 | -11.227 | 2.757 | .235 |
| Less than 1 serve a day | -3.680 | 3.4413 | -10.425 | 3.065 | .285 |
| 1 serve per day | -3.690 | 3.4211 | -10.395 | 3.015 | .281 |
| 2 serves per day | -3.573 | 3.4211 | -10.278 | 3.132 | .296 |
| 3 serves per day | -3.117 | 3.4090 | -9.798 | 3.565 | .361 |
| 4 serves per day | -4.282 | 3.6276 | -11.392 | 2.828 | .238 |
| 5 serves per day | -4.171 | 3.7866 | -11.593 | 3.250 | .271 |
| 6 or more serves per day | Ref | Ref | Ref | Ref | Ref |
| **Serves of vegetables per day** | | | | | |
| I never eat vegetables | -8.975 | 6.6932 | -22.093 | 4.143 | .180 |
| Less than 1 serve per day | -3.122 | 2.7877 | -8.586 | 2.342 | .263 |
| 1 serves per day | -3.125 | 2.6963 | -8.410 | 2.160 | .246 |
| 2 serves per day | -2.593 | 2.6668 | -7.820 | 2.634 | .331 |
| 3 serves per day | -4.409 | 2.6839 | -9.669 | .851 | .100 |
| 4 serves per day | -2.713 | 2.7389 | -8.081 | 2.655 | .322 |
| 5 serves per day | -3.580 | 2.9176 | -9.298 | 2.139 | .220 |
| 6 or more serves per day | Ref | Ref | Ref | Ref | Ref |
| **Cups of sugar sweetened beverages** | | | | | |
| Less than 1 cup per month | -.130 | .9397 | -1.972 | 1.711 | .890 |
| 1 cup per week | .382 | .9121 | -1.406 | 2.169 | .676 |
| 2–6 cups per week | .657 | .9265 | -1.159 | 2.473 | .478 |
| 1 cup per day | 1.020 | .9982 | -.937 | 2.976 | .307 |
| 2 cups per day | 1.121 | 1.0507 | -.939 | 3.180 | .286 |
| 3 cups per day | 2.672 | 1.1506 | .417 | 4.927 | .020* |
| 4 cups per day | 4.238 | 1.5257 | 1.247 | 7.228 | .005* |
| 5 cups per day | 3.966 | 2.0814 | -.113 | 8.046 | .057 |
| 6 cups per day | 2.491 | 1.9492 | -1.329 | 6.311 | .201 |
| 7 cups per day | 6.819 | 3.1786 | .589 | 13.049 | .032* |
| 8 cups per day | 3.210 | 3.3683 | -3.392 | 9.812 | .341 |
| More than 8 cups per day | 2.141 | 2.7545 | -3.257 | 7.540 | .437 |
| I never drink sugar sweetened beverages | Ref | Ref | Ref | Ref | Ref |
| **Fast food consumption** | | | | | |
| Less than 1 per month | 2.819 | 1.3310 | .211 | 5.428 | .034* |
| 1–3 times per week | 2.441 | 1.4250 | -.351 | 5.234 | .087 |
| Once per week | 3.471 | 1.3523 | .820 | 6.121 | .010* |
| 2 times per week | 3.204 | 1.4986 | .267 | 6.141 | .032* |
| 3 times per week | 2.067 | 1.7887 | -1.439 | 5.572 | .248 |
| 4 times per week | 6.450 | 2.1892 | 2.159 | 10.740 | .003* |
| 5 times per week | 4.043 | 2.5929 | -1.039 | 9.125 | .119 |
| 7 times per week | 3.331 | 3.1948 | -2.931 | 9.592 | .297 |
| More than 7 times per week | 11.108 | 5.9543 | -.562 | 22.779 | .062 |
| Never | Ref | Ref | Ref | Ref | Ref |
| **Discretionary potato consumption** | | | | | |
| Less than 1 per month | .555 | 1.3813 | -2.152 | 3.262 | .688 |
| 1–3 times per week | .376 | 1.4508 | -2.467 | 3.219 | .796 |

*(Continued)*

**Table 3.** (Continued)

| Parameter | Co-efficient | Std. Error | 95% Confidence Interval | | P |
| --- | --- | --- | --- | --- | --- |
| | | | Lower | Upper | |
| Once per week | .340 | 1.4112 | -2.425 | 3.106 | .809 |
| 2 times per week | -.378 | 1.6003 | -3.514 | 2.759 | .813 |
| 3 times per week | -.835 | 1.7809 | -4.325 | 2.656 | .639 |
| 4 times per week | -1.747 | 1.7107 | -5.099 | 1.606 | .307 |
| 5 times per week | -2.608 | 2.0536 | -6.633 | 1.417 | .204 |
| 6 times per week | 1.773 | 4.9648 | -7.958 | 11.504 | .721 |
| 7 times per week | 1.957 | 2.9207 | -3.768 | 7.681 | .503 |
| More than 7 times per week | -6.144 | 3.1355 | -12.289 | .002 | .050* |
| Never | Ref | Ref | Ref | Ref | Ref |
| **Sweet cake consumption** | | | | | |
| Less than 1 per month | -1.103 | .8587 | -2.786 | .580 | .199 |
| 1–3 times per week | -1.500 | .9320 | -3.327 | .326 | .107 |
| Once per week | -2.178 | .8757 | -3.895 | -.462 | .013* |
| 2 times per week | -2.087 | 1.0945 | -4.232 | .058 | .057 |
| 3 times per week | -2.421 | 1.4849 | -5.331 | .490 | .103 |
| 4 times per week | -.662 | 1.4436 | -3.491 | 2.168 | .647 |
| More than 5 times per week | -3.377 | 1.9966 | -7.291 | .536 | .091 |
| Never | Ref | Ref | Ref | Ref | Ref |

* Indicates significance at alpha level 0.05.

As 40% of study participants fell within the age category of 17–35 years (young adults), it ought to be highlighted that the propensity to gain weight is more pronounced during this life stage [32]. This is typically due to notable declines in physical activity [33] and increased consumption of discretionary foods [34]. As young adults transition out of the parental home and into full time employment, an increase in disposable income coupled with the disruption of pre-existing dietary habits and unfamiliar time pressures can often lead to poorer food habits and subsequent weight gain [35]. This weight gain is the highest rate compared with any other adult age group, equating to an annual increase of 0.5 to 1 kg from early to mid-adulthood [36, 37]. Factors such as higher salaries, increased time demands (shiftwork), access to discretionary foods and physical inactivity are all elements heavily representative in the mining industry (as our study highlighted). Worker's meal breaks are often spent in mess halls, where high calorie fast food is available on site, with access to drink vending machines dispensing SSB. The nature of the industry regarding shift work and long working hours often means workers report being too tired to cook healthy meals when at home and rely on convenience foods which are often high in calories [38]. This may offer one explanation of the increased rates of obesity within this population, that many workers are simply continuing with their trajectory of weight gain, albeit, accelerated due to the workplace factors highlighted above.

Within our sample, consumption of fruit and vegetables was considerably below national recommendations, and poorer than national general population figures [4]. Within Australia, the current recommendations for both adult men and women is the consumption of two serves of fruit per day and five serves of vegetables for women, whilst men aged 18–50 are recommended to have 6 serves of vegetables, and 5.5 serves for ages 51–70 [39]. Notwithstanding, participants in our study tracked similarly to general population data for fruit consumption (still below recommendations), however it must be stated that this is somewhat of an expected

**Table 4. Relationship between participant self-reported physical activity and sedentary behavior questionnaire and BMI.**

| Parameter | Co-efficient | Std. Error | 95% Confidence Interval | | P |
|---|---|---|---|---|---|
| | | | Lower | Upper | |
| **Total walking for fitness, recreation, or sport in the last week** | | | | | |
| 0 minutes | -.299 | 2.0584 | -4.333 | 3.735 | .885 |
| Between 1 and 149 minutes | 1.619 | 1.6116 | -1.539 | 4.778 | .315 |
| Between 150 and 300 minutes | .267 | 1.6816 | -3.029 | 3.563 | .874 |
| More than 300 minutes | Ref | Ref | Ref | Ref | Ref |
| **Total walking to get from place to place** | | | | | |
| 0 minutes | -1.983 | 2.1118 | -6.122 | 2.156 | .348 |
| Between 1 and 149 minutes | -1.310 | 1.3855 | -4.025 | 1.406 | .345 |
| Between 150 and 300 minutes | -1.785 | 1.6297 | -4.979 | 1.409 | .273 |
| More than 300 minutes | Ref | Ref | Ref | Ref | Ref |
| **Total amount of time you spent doing moderate exercise in the last week** | | | | | |
| 0 minutes | .401 | 2.2631 | -4.034 | 4.837 | .859 |
| Between 1 and 149 minutes | -1.186 | 1.8034 | -4.720 | 2.349 | .511 |
| Between 150 and 300 minutes | -.827 | 1.6514 | -4.064 | 2.409 | .616 |
| More than 300 minutes | Ref | Ref | Ref | Ref | Ref |
| **Total amount of vigorous exercise in the last week** | | | | | |
| 0 minutes | .566 | 2.1260 | -3.601 | 4.733 | .790 |
| Between 1 and 149 minutes | .559 | 1.9227 | -3.209 | 4.328 | .771 |
| Between 150 and 300 minutes | -2.889 | 1.9974 | -6.804 | 1.026 | .148 |
| More than 300 minutes | Ref | Ref | Ref | Ref | Ref |
| **Total amount of physical activity in the last week** | | | | | |
| 0 minutes | 1.651 | .9388 | -.189 | 3.491 | .079 |
| Between 1 and 149 minutes | .239 | .8258 | -1.380 | 1.857 | .773 |
| Between 150 and 300 minutes | 1.812 | .8535 | .139 | 3.485 | .034* |
| More than 300 minutes | Ref | Ref | Ref | Ref | Ref |
| **Total time sitting at work** | | | | | |
| 8 hours or more | 3.058 | 2.0049 | -.872 | 6.987 | .127 |
| 6 to 8 hours | 2.035 | 2.9648 | -3.776 | 7.846 | .492 |
| 4 to 6 hours | -1.657 | 2.6259 | -6.804 | 3.489 | .528 |
| 2 to 4 hours | 1.206 | 2.1915 | -3.089 | 5.501 | .582 |
| 1 minute to 2 hours | 1.505 | 2.0009 | -2.417 | 5.427 | .452 |
| No sitting | Ref | Ref | Ref | Ref | Ref |

* Indicates significance at alpha level 0.05.

**Table 5. Multiple linear regression model to predict BMI from shift work, fast food frequency and sugar sweetened beverage consumption.**

| Outcome | Unstandardized Coefficients | | Standardized Coefficients | t | Sig. | 95.0% Confidence Interval for B | |
|---|---|---|---|---|---|---|---|
| | B | Std. Error | Beta | | | Lower Bound | Upper Bound |
| **(Constant)** | 27.674 | .586 | | 47.263 | .000 | 26.525 | 28.824 |
| **Shift work** | 1.145 | .509 | .080 | 2.250 | .025* | .146 | 2.145 |
| **Fast food** | -.076 | .103 | -.026 | -.733 | .464 | -.278 | .127 |
| **SSB** | .119 | .065 | .065 | 1.824 | .069 | -.009 | .247 |

* Indicates significance at alpha level 0.05.

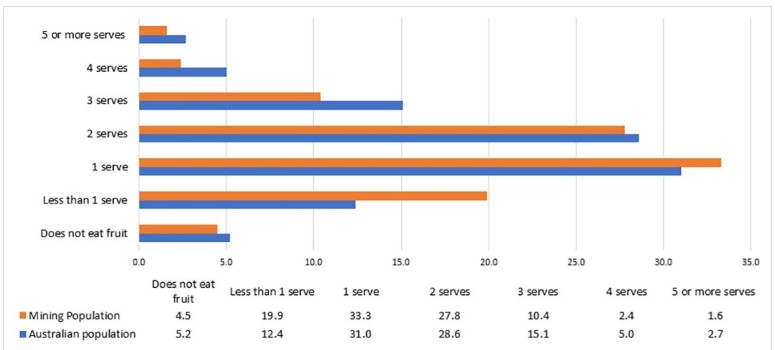

**Fig 1. Comparison of usual daily consumption of fruit per serve: Mining Population compared to General Australian Population.**

outcome. This is due in part to knowledge of recommendations and serving sizes pertaining to fruit and vegetables [40]. A study of 1108 participants by Pollard et al. [41], assessing knowledge of fruit and vegetable serving sizes in Western Australian adults highlights this phenomenon reporting that 42% of participants could correctly identify the serving size for fruit compared to 14% for vegetables. This may be due in part to the success of public health campaigns such as the *'Go for 2&5'* program [42], promoting the consumption of two pieces of fruit a day.

Vegetable consumption followed similar trends to fruit with inadequate intake reported. Considering the male dominated nature of participants, with 90% below the age of 54, consumption of the recommended six serves of vegetable was almost non-existent (1%), however it must be added that with regards to national figures for men only 3.2% are meeting this requirement. Knowledge is a big contributor to poor vegetable intake [41], although within male hegemonic masculinity literature, the feminization of vegetables and dieting as a whole can deter men from making healthy food choices [43]. This is even more represented in blue-collar male dominated industries such as manufacturing, construction and mining [44]. Knowledge plays a significant role in eating a healthy diet and these findings represent opportunities for possible future workplace health programs.

Habit formation is an effective behavior change technique for cultivating positive health behaviors [45]. However, within work settings which utilize shift work, it can be challenging for workers to have the consistent schedule to successfully implement lifestyle changes to

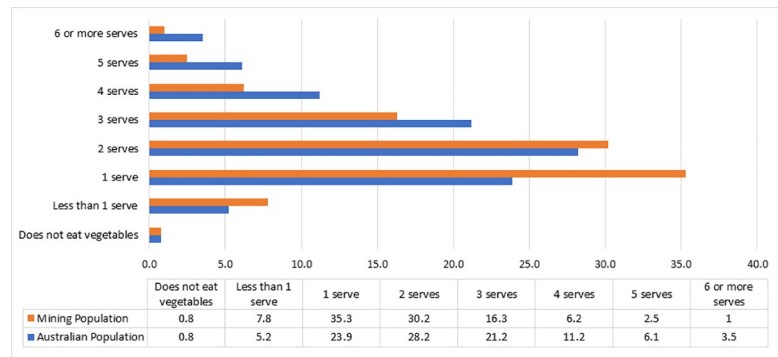

**Fig 2. Comparison of usual daily consumption of vegetables per serve: Mining Population compared to General Australian Population.**

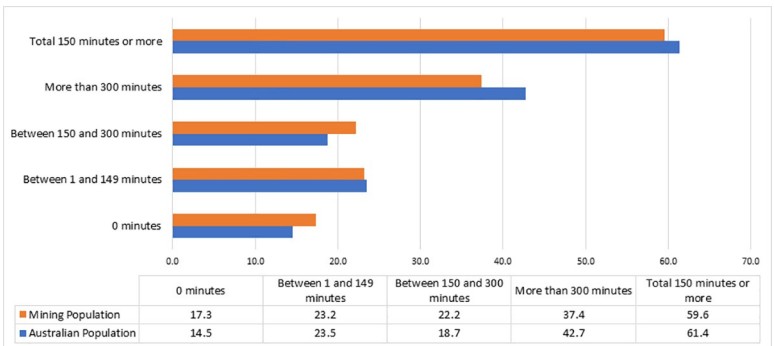

| | 0 minutes | Between 1 and 149 minutes | Between 150 and 300 minutes | More than 300 minutes | Total 150 minutes or more |
|---|---|---|---|---|---|
| Mining Population | 17.3 | 23.2 | 22.2 | 37.4 | 59.6 |
| Australian Population | 14.5 | 23.5 | 18.7 | 42.7 | 61.4 |

**Fig 3. Comparison of physical activity minutes per week: Mining Population compared to General Australian Population.**

promote healthier habits. In a different industry, but with similar shift work tendencies, a study by Heath et al., studied 40 nurses over 181 rotating shifts to compare energy intakes for different shifts [46]. Afternoon shifts were associated with a lower total daily energy intake ($\beta = -1659.4$, P<0.01) when compared to night or morning shifts [46]. It has been previously documented that shift workers' overall diet quality is poorer than their counter parts, with an increased reliance on nutrient poor, energy dense foods as well as increased snacking frequency [47]. Considering 80.1% of the participants in this study worked shifts, which is common practice in mining [13, 48], it is not surprising to see poorer health outcomes in study participants including elevated BMI, higher consumption of fast food and SSB, as well as inadequate intake of fruit and vegetables. With regards to elevated BMI and shift work status, the significant association within our study mirrors a sentiment which has been reported elsewhere in the literature [49–51], and more specifically to Australian men [52].

Physical activity plays an important role in weight homeostasis [53]. Interestingly within our sample, participants who belonged to the 150 to 300 minutes per week category of physical activity showed a significant relationship with increasing BMI. Whilst recommended levels of physical and increasing BMI is rather contradictory, this could be explained by sex preferences towards certain methods for weight loss. Within masculinity research, it has been shown that men have a greater affinity towards physical activity as opposed to dieting to illicit weight loss [43]. As the sample is 90% males, this may offer an explanation for this phenomenon; that men are increasing physical activity in response to excess weight, albeit, without success, as the consequential effects of a bad diet can at times outweigh increased exercise [54]. This is further reinforced within the literature, with studies reporting that physical activity alone has a diminished effect of weight balance (when compared to diet alone and a combination of diet and physical activity) [55, 56]. Additionally, the intensity of exercise rather than minutes active has been shown to be more influential with regards to weight maintenance [55]. Exploration of this using objective measures (e.g. accelerometer or heart rate trackers) would help affirm this notion.

Workplace health promotion programs, particularly amongst men, who utilized both diet and exercise components have shown success [57]. One example of a study which incorporated both these themes was the Preventing Obesity without Eating like a Rabbit (POWER) by Morgan et al. [58]. In a two-armed randomized control trail of 110 overweight or obese male employees at a metal manufacturing plant, significant changes (P <0.001) to weight (-4.3 kg), waist circumference (-5.9cm) and BMI (-1.4), were observed all favoring the intervention group. Using a multi-component intervention that focuses on both diet and physical activity

may produce a greater effect (with regards to weight outcomes) for employees participating in a workplace health promotion program.

The goal of RESHAPE however is not to implement a once-off wellness program, rather bring about organizational change and the cultivation of an environment which values the health of its employees. This long-term success does not simply depend on the contents of the program; instead the models used to action the intervention can implicate health outcomes at an individual level [59]. The RESHAPE framework takes a non-traditional approach to workplace wellness and looks at organizational change as opposed to a solely focusing on the individual [60]. This is evident by step three of the RESHAPE framework which looks to align the RESHAPE charter to organizational principles. This step is important for an organization as it mandates that health and wellness will be a part of the ethos of the establishment going forward, and clearly articulates to employees that their health is valued. Without this organization change it is unlikely that the workplaces will cultivate any meaningful change with regards to the health and wellness of their employees [61].

Despite the theoretical relevance of sustainably enacted wellness programs in the workplace, there exists a dearth of literature concerning this subject matter [61]. This may be attributed the degree of difficulty of continuously implementing programs that are effective. Jenny and colleagues [62] in their study concerning process and outcome evaluations of a workplace stress management program further ratify this notion; that the successful implementation of workplace programs requires several overarching themes including: perseverance, strong coalitions, constant fine-tuning and support, systematic training and reflection. Once again circling back to the necessity of a framework (compared to the contents of the program), RESHAPE incorporates all these aspects within its eight steps. Ultimately RESHAPE looks to apply these theories within the workplace for the good of the individual and organization as a collective whole.

## Limitations

All data in this study is self-reported, therefore there are inherit bias involved with this type of research. Recall and respondent biases concerning nutrition and physical activity responses may lead to under-reporting [63] which is a common issue with these types of survey tools. Height and weight were also self-reported, which provided the measurement outcomes to calculate BMI. Men and women typically slightly over-report height and under-report weight [64]. This may mean that our results are not truly reflective of study participants assessed, and some level of caution is prescribed when contextually evaluating these findings. Questions related to alcohol intake, mental health, anxiety, as well as workload and stresses may be somewhat of a sensitive issue amongst study participants, primarily due to fear of job loss. Therefore, caution is warranted when interrupting these results. This study did not report findings relating to workplace productivity, sleep, fatigue, and mental health. These findings were omitted as they did not suit the theme of the paper.

## Conclusion

As obesity is multi-factorial in nature, there are several underlying elements contributing to the occurrence of excessive weight within the NSW coal mining industry. Poor nutrition is likely to be a key component in this scenario, however, this is further compounded by workplace factors which exacerbate challenges workers face in implementing health behavior change (e.g., participation in shift work). Overall, this paper had a particular focus on diet and physical activity as the main outcomes, which will ultimately be the primary driver for the RESHAPE initiatives in achieving and maintaining a healthy weight for coal miners. Future

studies would benefit from assessing diet and physical activity knowledge in relation to recommendations and serving sizes. Multi-component interventions that holistically look at the issue and are championed within conducive work environments may have the greatest impact at both the individual and industry level.

## Acknowledgments

The Authors would like to acknowledge the contributions of all the workers who participated as well as Mrs. Emma Ford for aiding in data entry.

## Author Contributions

**Conceptualization:** Trent Watson, Carole L. James.

**Data curation:** Aaron Bezzina.

**Formal analysis:** Aaron Bezzina, Emma K. Austin.

**Funding acquisition:** Carole L. James.

**Investigation:** Aaron Bezzina, Emma K. Austin, Carole L. James.

**Methodology:** Aaron Bezzina, Trent Watson, Carole L. James.

**Project administration:** Emma K. Austin, Trent Watson, Carole L. James.

**Resources:** Emma K. Austin.

**Software:** Aaron Bezzina.

**Supervision:** Trent Watson, Lee Ashton, Carole L. James.

**Validation:** Aaron Bezzina.

**Visualization:** Aaron Bezzina, Emma K. Austin, Lee Ashton.

**Writing – original draft:** Aaron Bezzina, Emma K. Austin, Carole L. James.

**Writing – review & editing:** Aaron Bezzina, Emma K. Austin, Trent Watson, Lee Ashton, Carole L. James.

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
