## [Decision Letter · Decision Letter 0]

30 Mar 2021

PONE-D-21-05421

Health and wellness in the Australian coal mining industry: A cross sectional analysis of baseline findings from the RESHAPE workplace wellness program.

PLOS ONE

Dear Dr. Bezzina,

Thank you for submitting your manuscript to PLOS ONE. After careful consideration, we feel that it has merit but does not fully meet PLOS ONE’s publication criteria as it currently stands. Therefore, we invite you to submit a revised version of the manuscript that addresses the points raised during the review process.

We look forward to receiving your revised manuscript.

Kind regards,

Frank T. Spradley

Academic Editor

PLOS ONE

"This research was supported by funding from Coal Services Health and Safety Trust

G1800980."

We note that one or more of the authors are employed by a commercial company: "Ethos Health,"

and

We note that you received funding from a commercial source: "Coal Services Health and Safety Trust"

b) Please also provide an updated Competing Interests Statement declaring this commercial affiliation and funder along with any other relevant declarations relating to employment, consultancy, patents, products in development, or marketed products, etc. 

4. We note you have included a table to which you do not refer in the text of your manuscript. Please ensure that you refer to Table 1 in your text; if accepted, production will need this reference to link the reader to the Table.

Reviewers' comments:

Reviewer's Responses to Questions

**Comments to the Author**

1. Is the manuscript technically sound, and do the data support the conclusions?

Reviewer #1: Yes

2. Has the statistical analysis been performed appropriately and rigorously? 

Reviewer #1: I Don't Know

3. Have the authors made all data underlying the findings in their manuscript fully available?

Reviewer #1: No

4. Is the manuscript presented in an intelligible fashion and written in standard English?

Reviewer #1: Yes

5. Review Comments to the Author

Reviewer #1: 

1. I know space is limited, but a bit more information about the RESHAPE program would help the abstract.

2. The phrase “5 serves of vegetables” in the abstract needs to be corrected.

3. What is novel about this study. Any similar studies in a comparative cohort from other countries? And what about this specific industry linked to greater rates of overweight/obesity versus other labor industries?

4. The introduction is too long and should focus on the rationale for this study – it should be 2 pages at most. Instead providing “baseline data” as a rationale for conducting this study, a directional hypothesis statement needs to indicate what is expected about diet and physical activity programs on rates of overweight and obesity in blue-collar workers.

5. As mentioned in the introduction, is there any specific information about the workplace health promotion programs? How does this study help overcome time and knowledge and gender targeted interventions to implement workplace health programs?

6. In the conclusions, it is stated, “Poor nutrition is likely to be a key component in this scenario; however, this is further compounded by workplace factors which exacerbate challenges workers face in implementing health behavior change.” What are these factors, and can they be assessed here?

7. RESHAPE is not mentioned much in the discussion.

8. Why not include figures about physical activity?

9. Including a biostatistician would be helpful and provide rigor.

10. As funding was provided from a coal industry, is this some type of conflict of interest?

6. PLOS authors have the option to publish the peer review history of their article (what does this mean?). If published, this will include your full peer review and any attached files.

Reviewer #1: No

---

## [Author Response · Author response to Decision Letter 0]

12 May 2021

Please see attached the response to reviewer letter.

---

## [Decision Letter · Decision Letter 1]

24 May 2021

Health and wellness in the Australian coal mining industry: A cross sectional analysis of baseline findings from the RESHAPE workplace wellness program.

PONE-D-21-05421R1

Dear Dr. Bezzina,

We’re pleased to inform you that your manuscript has been judged scientifically suitable for publication and will be formally accepted for publication once it meets all outstanding technical requirements.

Kind regards,

Frank T. Spradley

Academic Editor

PLOS ONE

Reviewers' comments:

Reviewer's Responses to Questions

**Comments to the Author**

1. If the authors have adequately addressed your comments raised in a previous round of review and you feel that this manuscript is now acceptable for publication, you may indicate that here to bypass the “Comments to the Author” section, enter your conflict of interest statement in the “Confidential to Editor” section, and submit your "Accept" recommendation.

Reviewer #1: All comments have been addressed

2. Is the manuscript technically sound, and do the data support the conclusions?

Reviewer #1: Yes

3. Has the statistical analysis been performed appropriately and rigorously? 

Reviewer #1: Yes

4. Have the authors made all data underlying the findings in their manuscript fully available?

Reviewer #1: Yes

5. Is the manuscript presented in an intelligible fashion and written in standard English?

Reviewer #1: Yes

6. Review Comments to the Author

Reviewer #1: The authors have adequately addressed all previous comments and this manuscript is suitable for publication.

7. PLOS authors have the option to publish the peer review history of their article (what does this mean?). If published, this will include your full peer review and any attached files.

Reviewer #1: No

---

## [Editor Report · Acceptance letter]

28 May 2021

PONE-D-21-05421R1 

Health and wellness in the Australian coal mining industry: A cross sectional analysis of baseline findings from the RESHAPE workplace wellness program. 

Dear Dr. Bezzina:

I'm pleased to inform you that your manuscript has been deemed suitable for publication in PLOS ONE. Congratulations! Your manuscript is now with our production department. 

Kind regards, 

on behalf of

Dr. Frank T. Spradley 

Academic Editor

PLOS ONE